# A Preliminary Scoping Review of the Impact of e-Prescribing on Pharmacists in Community Pharmacies

**DOI:** 10.3390/healthcare12131280

**Published:** 2024-06-26

**Authors:** Amr A. Farghali, Elizabeth M. Borycki

**Affiliations:** School of Health Information Science, University of Victoria, Victoria, BC V8P 5C2, Canada

**Keywords:** electronic prescribing, medication error, community pharmacists, pharmacy efficiency, patient safety

## Abstract

Objective: This scoping review aims to map the available literature and provide an overview of the published articles discussing the impact of electronic prescribing on medication errors and pharmacy workflow. Methods: The literature search was conducted using PubMed^®^, Web of Science^®^, and the Cochrane Database of Systematic Reviews^®^, as well as grey literature reports, using the search terms and related components of “pharmacists”, “electronic prescribing”, “medication errors”, and “efficiency”. The search included all articles that were published from January 2011 to September 2023. Twenty-two relevant articles were identified and fully reviewed, ten of which were included in this review. Results: Electronic prescribing (e-prescribing) provides a solution for some of the challenges that are associated with handwritten and paper prescriptions. However, the implementation of e-prescribing systems has been recognized as a source of new unforeseen medication errors in all the reviewed articles. Productivity in community pharmacies has been affected with receiving electronic prescriptions (e-prescriptions) and having to deal with the issues that arise from them. The pharmacists’ interventions were not eliminated with e-prescriptions compared to other prescription formats. The most frequently reported reason for intervention was related to incomplete instructions in the field of directions of use. Other common challenges with e-prescriptions were related to missing information, quantity, inappropriate dose, dosage form, and drug. Discussion: This review demonstrates the scarcity of research about the impact of electronic prescribing on medication error and efficiency in community pharmacies. In the literature, most of the studies had mainly focused on hospital pharmacies. The literature search demonstrated that there are still some barriers to overcome with e-prescribing systems and that medication errors were not fully eliminated with e-prescriptions. New errors have been identified with e-prescriptions, all of which caused delays in processing, which affected the productivity of the pharmacy staff, and could have negatively impacted patients’ safety if not properly resolved. Conclusion: e-Prescribing solved some of the challenges associated with illegibility of handwritten prescriptions. However, more time is required to allow e-prescribing systems to mature. Further training for prescribers and pharmacists is also recommended before and after the implementation.

## 1. Introduction

Electronic prescribing (e-prescribing) systems are intended to establish a direct communication between the prescriber and pharmacy computer systems [1,2]. One of the major benefits of e-prescribing is the elimination of medication errors due to illegible handwriting [3]. There has been a growing interest in adopting and implementing e-prescribing systems around the world to improve patient safety and enhance the provision of healthcare. e-Prescribing is described as a solution that improves the quality of care, helps with reducing physician and pharmacist errors, and helps to decrease malpractice claims [4].

Several studies suggest that e-prescribing can reduce prescribing errors, and increase efficiency [5,6,7]. The definition of prescribing errors has been largely debated. Aronson defined a prescription error as ‘a failure in the prescription writing process that results in a wrong instruction about one or more of the normal features of a prescription’ [8]. The ‘normal features’ include the identity of the recipient, the identity of the drug, the formulation, dose, route, timing, frequency, and duration of administration [8].

e-Prescribing can be defined as the creation of a digital prescription and securely transmitting it from the prescriber’s electronic medical record (EMR) system directly to the pharmacy management system (PMS) through a secure, direct communication [6]. Electronic prescriptions aim to replace all other formats of prescriptions, including handwritten and EMR-generated prescriptions. Eliminating the need to manually enter and transcribe prescriptions into the PMS will reduce the chance pharmacy-induced errors, including transcribing errors [9].

According to a systematic review of e-prescribing practices in the ambulatory care setting published about a decade ago, e-prescribing reduces prescribing errors, increases efficiency, and helps to save on healthcare costs [10]. The focus of this review has been directed to community pharmacies since they represent the majority of pharmacy practice in Canada [11]. The main difference between a community and a hospital pharmacy is the type of patients they both serve. A community pharmacy’s main services are directed to the general public, while a hospital pharmacy is responsible for overseeing the medication management process for inpatients and outpatients in a hospital setting.

## 2. Study Rationale

Innovative health information technologies have been widely embraced by pharmacists to assist in their patient care duties. However, we need to acknowledge that changes in technology can also have an influence on the nature of a pharmacy’s workflow and the pace of pharmacist work, whether in hospital or in community practice.

For that reason, awareness needs to be raised about the impact of technology on a pharmacy working environment in terms of efficiency and productivity in the workplace [12]. Most importantly, there is a need to assess the impact of introducing these new technologies on patients’ safety and health outcomes. To address this gap in the literature, a scoping review was conducted to accumulate knowledge and learn about the current state of the literature.

## 3. Study Objective

This scoping review aims to map the available literature and provide an overview of the published articles discussing the impact of electronic prescribing on medication errors and pharmacy workflow. The objective is to provide an overview of the community pharmacists’ perception of the impact of electronic prescribing on pharmacist work, workflow, medication errors, productivity, and patient outcomes in community pharmacies.

## 4. Methods

This scoping review followed an enhanced Arksey and O’Malley methodological framework as described by Daudt et al. [13]. This enhanced framework describes the five steps of a scoping review as follows: identifying the research question; identifying relevant studies; study selection; charting the data; and collating, summarizing, and reporting the results.

### 4.1. Identifying the Research Questions

Based on the initial search conducted before writing this review, limited research was found addressing community pharmacies in general. Therefore, the research questions for this scoping review have been determined to focus on the following:What are the community pharmacists’ perceptions of the impact of electronic prescribing on medication errors and patient outcomes in community pharmacies?What are the community pharmacists’ perceptions of the impact of electronic prescribing on the pharmacist work and productivity in their pharmacy?

### 4.2. Identifying Relevant Studies

Identification of the relevant literature, whether qualitative or quantitative studies, was performed by conducting an initial search using two electronic databases (i.e., PubMed^®^ and Web of Science^®^). The Cochrane Database of Systematic Reviews^®^ was also reviewed in a second search to identify any systematic or scoping reviews pertinent to the research questions. To ensure that all relevant information was captured, the Grey Literature Report, an online grey literature database repository of reports, working publications, and conference proceedings, was also searched. 

The search for this review was performed for published literature from January 2011 to September 2023 and was limited to English language articles only. A preliminary search of the literature was completed before conducting this review, and most of the articles relevant to our review questions started to be published in 2011. For that reason, it was decided that the search dates will include articles published between 2011 and 2023. The limitation of English only articles was imposed due to the inability to secure a reliable source of translation from other languages into English.

An experienced research librarian was consulted before starting the search to ensure that the selection of databases, search terms, and search strategy was thorough. Searches were performed using the search terms in Table 1.

## 5. Study Selection

The inclusion criteria of this scoping review are listed in Table 2. These criteria were tested on a few of the identified articles to ensure that they are capturing all articles related to the research questions. The first search was performed using PubMed^®^ and Web of Science^®^ databases and included two queries. The date limitation was set from January 2011 to September 2023, during the study selection process. Firstly, we used all fields to search for all articles about e-prescribing, pharmacists, and medication errors. Then the second query was used to search for articles about e-prescribing, pharmacists, and efficiency. This step was added to ensure we captured all relevant articles that might be related to our second research question. The Cochrane Database of Systematic Reviews and the Grey Literature Report were also searched for relevant articles.

During the first stage of screening, a review of titles and abstracts was conducted by two researchers to select all relevant articles (AF and EB). A full-text examination was performed at the second stage of screening. Relevant studies were included only if they were found to fit the inclusion criteria (see Table 2).

### 5.1. Charting the Data

An iterative process was used to classify the general themes extracted from the selected articles in the review. This process was initially started by reading each article individually to identify the category of evidence presented in each article. Then the charting of the data was completed using Microsoft Excel. The charting form was used to collect some descriptive characteristics of the selected articles, such as the main author, year of publication, and country of origin. A data summary of the characteristics of the articles is provided in Table 3.

### 5.2. Collating, Summarizing, and Reporting the Results

Scoping reviews are intended to provide an overview of the characteristics or factors related to a particular concept regardless of the quality of evidence [23]. For that reason, no critical appraisal was conducted to evaluate the quality of evidence in the selected articles. The variables sought in the articles were divided into two main categories: e-prescribing-related variables influencing medication errors (Table 4), and e-prescribing-related variables affecting productivity in the pharmacy (Table 5). Synthesis of the results and the data were collated and summarized across all the articles following these two categories and findings presented accordingly.

## 6. Results

### 6.1. Study Selection

The searches in PubMed^®^ and Web of Science^®^ databases were performed in September 2023. The first query resulted in 158 articles and the second one resulted in 82 articles. On searching the other databases, 30 additional articles were found on Cochrane Database of Systematic Reviews^®^ and no results were found in the Grey Literature Report^®^. After removal of duplicates, a total of 199 articles were included. The first screening process was performed by screening the titles and abstracts of all the identified articles by two researchers (EB and AF). Only 24 articles met the inclusion criteria. These articles were screened further by conducting a full text reading. The second screening yielded 10 relevant articles. Figure 1 illustrates the study selection process.

### 6.2. Study Characteristics

Most of the articles in this review, around 60% (n = 6) originated from the United States, while the others were from Finland (n = 2), Canada (n = 1), and Sweden (n = 1). The oldest article identified was published in 2011 [22], which indicates that less research has been conducted on community pharmacies while the focus of most of the literature remains on hospital pharmacies.

### 6.3. Characteristics of the Subjects in the Studies 

All subjects in this review were healthcare professionals practicing in a community pharmacy setting; whether as pharmacists, pharmacy students or pharmacy technicians and using e-prescribing systems. However, one study examined community pharmacies that did not accept e-prescriptions [18]. This study aimed to explore the barriers to adopting systems from a pharmacist’ perspective as they impact the pharmacist’s decision to adopt e-prescribing systems (i.e., whether the pharmacists were planning on using the e-prescribing systems in the future or not).

### 6.4. Country and Pharmacy Characteristics

The definition of e-prescribing varied between different countries. Most of the e-prescribing systems used in this review included prescriptions that were electronically generated by a prescribers’ office and securely transmitted electronically (i.e., directly) to the pharmacy’s computer system. Pharmacies that printed e-prescriptions and dealt with them as paper prescriptions were excluded [21].

### 6.5. Data Collection Technique

Many of the articles in this review applied observational study designs and used direct observation of the dispensing process in the participating pharmacies [14,15,19,21]. Postal questionnaires were used to survey pharmacists’ perceptions of e-prescribing systems and their thoughts about their systems’ strengths and pitfalls [17,22]. Two studies involved a secondary analysis of pre-existing datasets. One was on the National Survey of Community-Based Pharmacists [6], and the other one was on the Dyke Anderson Patient Safety Database (DAPSD) [16]. The National Survey of Community-Based Pharmacists data was collected using a web-based survey sent to community pharmacists in Canada by email [6]. The DAPSD dataset was collected using surveys of pharmacists working in different pharmacy settings in the state of Nebraska [24]. Telephone interviews were also employed, and the interviews employed both structured and semi-structured interview formats [18,20].

### 6.6. e-Prescribing Impact on Medication Errors

To assess the impact of e-prescribing on medication errors, some variables (for example, inaccurate dose, inappropriate quantity, wrong drug) were captured and compared between the studies included in this review (Table 4). These variables were, for the most part, consistently collected across the differing articles. Once recognized in an e-prescription, all these variables might have led to a medication error, a potential risk to patients’ safety, and/or led to the disruption of the dispensing process. A pharmacist intervention was required to resolve the problem and to complete the medication dispensing process. 

The most common e-prescribing errors that required a pharmacist’s intervention had as a root cause an inaccuracy in the instructions field in the e-prescribing system regarding how to use the medication. The lack of clarity regarding the directions of medication use could lead to medication errors. Here, pharmacy staff or patients could potentially misunderstand the directions for taking the medication [16,17,20,21]. Another potential risk involving e-prescriptions was the use of abbreviation terms by prescribers in the instructions field instead of the full term(s), which can easily get misinterpreted or missed during transcription [17,19,20].

A potential source for medication errors was the use of additional field notes for extending the directions of use for a prescription. Those notes might not match the directions in the instructions field of the e-prescription which could lead to misinterpretation by the pharmacist or to the information being easily missed by pharmacy personnel [20].

Another challenge identified with e-prescriptions was the inability to accurately specify the quantity of the drug to be dispensed when the prescription is received by the pharmacy. Wrong drug quantity, duration, or day supply were reported as a regular occurrence in e-prescriptions which could also occur due to discrepancies between the directions of use and the requested quantity [17,19,21]. Other problems with e-prescriptions that increased the risk of medication errors included inappropriate drug choice, medication name, drug strength, or drug–drug interactions that were detected before the dispensing of medications [15,16,19,20]. Wrong drug selection has been reported in most of the articles, which could lead to serious clinical problems or drug–drug interactions.

The design of e-prescribing systems had an impact on medication safety as well. Some design challenges affected the proper selection of the appropriate medication (for example the drop-down menus), or the medication name was too long so the prescriber could not verify their proper selection because of it [17,21]. Other design issues were caused by inconsistencies between the size of the text fields between the prescriber’s and the pharmacy systems that could lead to data entry errors or missing some parts of the instructions of medication use [16,20,21]. Prescribing compounded medications using e-prescribing systems is a challenge due to the inconsistencies in drug names and interoperability issues between both computer systems [17].

Incorrect dose selection is another commonly occurring problem which could have a direct impact on patients’ safety if left undetected by the pharmacist [17,19]. Medication errors could occur due to the inaccurate selection of the medication dosage form which is a common challenge with e-prescriptions [17].

### 6.7. e-Prescribing Impact on Productivity

e-Prescribing systems have a direct impact on the workflow of all the pharmacies according to all the articles in this review (Table 5). The pharmacists described e-prescribing systems as more efficient for handling and processing prescriptions than the traditional paper format [6]. The average time for dispensing medications has decreased as less time was needed to spend on clarifying ambiguities of paper prescriptions due to illegible handwriting [14,22]. The process of retrieving, filing, and archiving the original prescription was also more efficient and faster for e-prescriptions. Another relevant feature of e-prescriptions was the ability to control medication hoarding that occurred during the COVID-19 pandemic [14].

However, there are still some of barriers to overcome with e-prescribing systems impacting the productivity of pharmacy staff while dispensing e-prescriptions [18]. When received in the pharmacy system, the pharmacy staff was often required to verify and manually change the instructions field of e-prescriptions before processing them and printing the pharmacy labels [17,20]. The manual selection of the prescribed medication was also necessary due to discrepancies between the prescriber and the pharmacy database systems [15,21]. A design challenge in e-prescribing systems was related to the excessive amount of information on e-prescriptions, which increased the time needed by the pharmacy staff to process them [21].

One study reported missing some legally required information related to the dispensing of controlled substances [19]. Missing information on the e-prescriptions led to delays, patients’ dissatisfaction, and pharmacists spending more time on solving the problem [17]. One of the important features of e-prescribing systems is the electronic refill option for renewing prescriptions, which allows for direct, prompt communication between the pharmacist and the prescriber [6]. However, the electronic renewal process was not optimally used by some pharmacies to avoid the associated service charges, or due to missing this function in the e-prescribing system [20].

Incidents of missing prescriptions occurred when patients were not able to receive their medications because of delays in sending the e-prescription or sending it to an incorrect pharmacy [20]. Missing prescriptions led to interruptions of the pharmacy workflow and more time spent by pharmacy staff to resolve the issue, which added to patients’ dissatisfaction. Pharmacy staff workload increased when duplicate orders were received, for example sending the same e-prescription by fax as well [16]. Delays with e-prescriptions also happened when the patient’s or prescriber’s information was not properly identified in the pharmacy system [15,21]. The pharmacy staff had to spend more time during the verification process to match the e-prescription with the right patient’s profile or create a new record for the prescriber if not found in the system.

Other barriers that affected productivity in the pharmacy, while dispensing e-prescriptions, were the insufficient staff training, computer and network connectivity problems, and inadequate support or maintenance [18,20,22]. Some pharmacies took a business decision and decided not to use e-prescribing systems due to the initial setup cost, or to avoid the transaction fees of e-prescriptions [18,20]. Some pharmacists thought that their intervention rates did not change with e-prescriptions compared to a handwritten or paper format [19].

## 7. Discussion

Research on the impact of technology on the workflow and medication safety in community pharmacies is scarce. The number of publications in hospital pharmacies and primary care settings continues to outnumber those in community pharmacy practice, even though the number of community pharmacies exceeds by far the number in any other pharmacy setting. Improving patients’ safety and reducing medication errors are two of the major attributes of e-prescribing systems that are being used to advocate for their widespread use in many countries [4,10]. The results of this review suggest that there are still some barriers to overcome with the adoption and full use of e-prescribing systems in retail pharmacies. 

The literature suggests that e-prescribing helps reduce the risk of medication errors that contribute to the poor legibility of handwritten prescriptions, decreases the turnaround time for refill requests, and reduces call-backs for unsigned prescriptions [17,20,25]. However, the findings of this review demonstrated that e-prescriptions might inadvertently introduce some unforeseen challenges that could lead to medication errors and impact the productivity in community pharmacies. And for that reason, adoption and implementation rates of e-prescribing in retail pharmacies are still not meeting expectations in many countries, for example in Canada [6,26].

### New Challenges with e-Prescribing

All the articles in this review demonstrated that there are some problems to overcome with e-prescriptions. The pharmacists’ interventions for medication errors were not fully eliminated with e-prescribing systems as it has been reported that there was no change in the intervention rates compared to the traditional paper prescriptions [19]. Most of the studies reported that the directions and quantity fields of the e-prescriptions were either incomplete or inappropriate. This finding is consistent with the evidence found in similar studies [27,28].

Medication errors occurred when prescribers used some Latin abbreviations in the instructions field that would not be understood by patients, for example, ‘b.i.d.’, which has to be manually changed to ‘take one tablet orally twice a day’ [17,20]. Some limitations in the e-prescription design forced the prescribers to choose unintended structured directions from a drop-down menu. As a result, the prescriber would use the additional notes field that allowed for free-text input to expand or elaborate on the directions provided in the structured field. However, using the additional notes field often did not match the provided directions in the instructions field in the e-prescription. This led to pharmacist confusion and required additional pharmacist time to clarify the e-prescriptions [20]. Sometimes these notes were just missed by the pharmacy staff, which could impose a potential risk for preventable medication errors.

This scoping review provided evidence that pharmacy staff were required to spend more time verifying and manually changing various fields of the e-prescription before dispensing a medication. Other studies reported similar findings indicating that e-prescriptions required more clarifications and time to resolve problems compared to other types of prescriptions [25,29,30,31]. The articles in this review indicated that more time was spent on resolving issues with e-prescriptions due to missing information, such as legal notation, quantity, or dose [17,19]. Manual drug selection was still required to be performed by the pharmacy staff due to discrepancies between the PMS and the EMR databases [15,20]. Depending on the training and experience of the staff, who were tasked e-prescriptions data entry, having to manually select the medication might lead to potential errors that could affect a patients’ safety.

Several researchers found patients were frustrated with e-prescriptions. Researchers identified that patients expect to pick up their medications when they arrive at the pharmacy. However, in many cases, the e-prescription had not been sent by the prescriber. Other studies had similar findings [25,32,33]. Delays in sending e-prescriptions arose from technical difficulties associated with the service, inadequate training on the prescriber or pharmacist side, or unintentionally sending of the e-prescription to the wrong pharmacy [20,25,29]. To resolve this problem, the researchers noted that pharmacists had to stop the task they were doing and had to contact the prescriber’s office to obtain a confirmation of the missing prescription, verbally or by fax. This activity caused interruptions in the pharmacy workflow and increased wait times for medications. Delays also happened when controlled substances (e.g., benzodiazepines, amphetamines, cannabinoids) were prescribed electronically. In some cases, the researchers found that the e-prescription was not compliant with the legal regulations in certain country jurisdictions [19,29].

Interoperability challenges between the EMR and the PMS created additional work for pharmacy staff. Mismatches happened due to differences in pack sizes or unit doses between the two systems that had to be corrected before processing an e-prescription. For example, when the prescribed medication is commercially available in a prepackaged form, such as an inhaler of 200 doses, or prefilled syringe of 3 mL, prescribers usually indicate in the quantity field the number of packs (e.g., three inhalers instead of 600 doses, or three syringes instead of 9 mL) [19,20]. These discrepancies cause additional processing time of e-prescriptions, leading to delays for the patients.

Wrong dose, strength, quantity, duration, and dosage form were common occurrences with e-prescriptions that could lead to significant medication errors, if not caught and addressed by a pharmacist. Pharmacists have also reported duplicate orders. Duplicate orders arise, when an e-prescription is sent, and the same prescription is also sent by other means (such as a fax or over the phone) by a prescriber. This negatively impacts the workflow in the pharmacies (i.e., pharmacists need to check for duplicate orders, and, in some cases, they need to contact the prescriber to confirm the order).

Other new challenges with e-prescribing systems included resistance from staff and the need for ongoing technical support [34]. Alert fatigue and a need for continuous updates to the database system to include new drugs, drug interactions, and contraindications has been reported [35]. Some pharmacies have decided to delay the implementation of e-prescribing systems due to business concerns. Pharmacies have identified e-prescribing start-up costs and the cost of paying transaction fees for e-prescriptions as reasons for delaying the implementation of this technology [18,20].

## 8. Limitations

This scoping review is first attempt to provide evidence of and highlight the impact of e-prescribing systems upon community pharmacies. To the best of our knowledge, no other reviews focused on community pharmacy workflow and error in the context of e-prescribing systems. One of the strengths in this review is the inclusion of articles from differing countries. This helped to identify common e-prescribing challenges experienced by varying nations globally, allowing for learning across healthcare systems. The review of differing country studies highlights the generalizability of the findings described in this review.

The review has a few limitations that must be acknowledged. Scoping reviews do not assess the quality of the evidence. Some of the articles in our review relied on self-reported tools to identify pharmacists’ perceptions about issues related to e-prescribing. The small sample size and limited duration of data collection period for the studies has also been identified as a limitation in some of the articles. A critical appraisal of the evidence presented in the articles was not performed as the intention of this review was to present available research regardless of the quality of evidence. 

The articles included in this review were published between 2011 and 2023, and the search was limited to two main databases PubMed^®^ and Web of Science^®^, along with the Grey literature. The Cochrane Database of Systematic Reviews^®^ was also reviewed for any other systematic or scoping reviews on our topic. The articles included in the review were also limited to English language only. More articles dated before 2011, after 2023, and other literature sources might have led to more articles that would fit the inclusion criteria and might have had valuable insights. Lastly, in reviewing articles from varying countries, it was noted that the implementations of e-prescribing services varied, and this may have had an impact on the pharmacy workflow and medical error outcomes.

## 9. Conclusions

This scoping review provided evidence about some of the inadvertent challenges created due to the introduction of e-prescribing in primary care and community pharmacies. Medications errors and workflow interruptions are still occurring when prescriptions are received electronically. No decline was noticed in the rate of pharmacist’s interventions dealing with ambiguities related to e-prescriptions. The time spent by pharmacy staff resolving issues related to e-prescriptions was not significantly different than other formats of prescriptions (i.e., paper-based prescriptions, faxed prescriptions). 

The findings of this review indicate that there are potential benefits for implementing e-prescribing systems in primary care and community pharmacies. However, system design improvements and interoperability issues need to be addressed to ensure efficiency and seamless transmission of prescriptions between the prescriber and the pharmacy before reaching the patient. More research is needed to understand the effects of introducing e-prescribing on pharmacist work. This could include more investigation into solutions that would help to address the challenges identified in this review.

## Figures and Tables

**Figure 1 healthcare-12-01280-f001:**
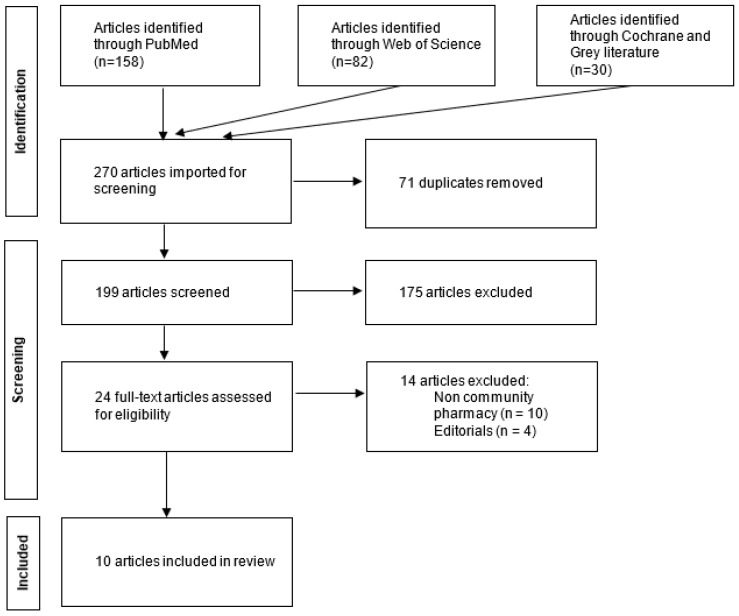
Flow diagram for the study selection process.

**Table 1 healthcare-12-01280-t001:** Key word search terms.

Key Word Search Terms (Synonyms Using OR)	AND
Search#1 “Electronic prescribing” OR “e-prescribing” OR “ePrescribing”	AND “Pharmacist” OR “Pharmacy” AND “Medication error” OR “Medical error”
Search#2 “Electronic prescribing” OR “e-prescribing” OR “ePrescribing”	AND “Pharmacist” OR “Pharmacy” AND “Efficiency” OR “Effectiveness” OR “Productivity”

**Table 2 healthcare-12-01280-t002:** Inclusion and exclusion criteria.

Inclusion Criteria	Exclusion Criteria
- Electronic prescribing - Community pharmacists - Compounding pharmacists - Retail pharmacists - Medication errors related to e-prescribing - Workflow, efficiency, or productivity analysis study - Primary care practices	- Editorials - Non-English articles - Articles not available as full text - Hospital systems - Medication reconciliation systems - Literature reviews - Systematic reviews - Protocol alone

**Table 3 healthcare-12-01280-t003:** Data summary of articles included in the review.

First Author	Title	Objective	Participants	Study Design	Data Collections	Results	Conclusion
Peltoniemi [14]	Electronic prescription as a driver for digitalization in Finnish pharmacies	Studying e-prescription systems in terms of their impact on workflows and practices in pharmacy using a sociotechnical perspective. The objective is to understand the disruption caused by e-prescribing systems to the pharmacy staff and how time management has changed due to their implementation.	Five mid-size community pharmacies in the urban area of Helsinki.	Observational retrospective study.	Direct observation of the dispensing process from distance from the pharmacy counter.	A decrease in median and average time was identified for the delivery times of the different types of prescriptions in the Finnish community pharmacies. In 2006, the average time for prescription dispensing was 3 min 26 s, compared to 1 min 58 s in 2012. The average dispensing time for a single prescription was 13% shorter using e-prescribing systems in 2012. The minimum and maximum times were also reduced. The maximum time for dispensing paper prescriptions in 2006 was 12 min and 40 s, which has been reduced to 6 min 22 s.	The implementation of e-prescribing systems reduced the processing and dispensing time of prescription medications in the community pharmacies in Finland. The dispensing process has become more digitized and less depending on human skills.
Panich [15]	Assessing automated product selection success rates in transmissions between electronic prescribing and community pharmacy platforms	Measuring the success rate of e-prescribing systems’ automatic product selection in community pharmacies. The aim is to identify the reasons leading to the failure of the automated process. Another goal of the study is to evaluate the accuracy of prescription dispensing, before and after the implementation of e-prescribing.	Direct observation in 2 outpatient pharmacies, and 14 community pharmacies participated in the surveys.	Observational study design	Direct observation led by a pharmacist or a pharmacy technician and mail surveys.	Automated product selection failure was detected in the sample data collected that consisted of almost 900 e-prescriptions. A total of 20.3% of e-prescription automated product selection failed, including failed drug selection due to mismatched NDC (60%), prescriber not identified (26%), and patient not found (14%). Surveys received from other pharmacies receiving e-prescriptions indicated that failure of the automated product selection occurred 10%–49% of the time.	
Shah [16]	Error types with use of medication-related technology: A mixed methods research study	Evaluation of the new type of errors that triggered by new health information technologies, like e-prescribing systems and automated dispensing cabinets. The assessment of the specific type of errors was targeted before and after the integration of these technologies into the pharmacy workflow.	A pre-existing dataset was used that involved pharmacists practicing in all pharmacy settings across the state of Nebraska.	A sequential transformative mixed method.	Surveys of 535 pharmacists working in different pharmacy settings in Nebraska was the method used in the original dataset, which was reused for this study.	e-Prescription use in pharmacies led to the elimination of some error types, including illegible handwriting of paper prescriptions and transfer of incorrect patient information. Three types of errors persisted after e-prescriptions use that included time delays during the prescribing or dispensing process, inaccuracy in drug regimen (wrong drug, wrong dose, wrong frequency, wrong route, or wrong dosage form), lack of information to safely process prescriptions, and wrong patient identification by the prescriber or pharmacist. Three new errors have emerged including computer system errors (risk-prone design), input error (wrong entry), and duplicate orders.	The use of health information technologies including e-prescribing have eliminated some common errors in pharmacy practice. However, these technologies were also the source of new types of errors that emerged after their incorporation in pharmacies’ daily routines. New strategies are still needed to reduce the risk of medication errors with the current e-prescribing systems that requires more future research.
Kauppinen [17]	The impact of electronic prescriptions on medication safety in Finnish community pharmacies: A survey of pharmacists	The main objective was to assess the pharmacists’ perspective of e-prescribing and its impact on medication errors in the community pharmacies. The study also explored the pharmacists’ thoughts on the frequency and different types of errors that they encountered with e-prescriptions.	A randomly selected sample of a total of 778 pharmacists from the community pharmacies.	A cross-sectional study design.	A four-page questionnaire that was mailed to a random sample of 1232 pharmacists.	Most of the pharmacists thought that e-prescribing improved the quality of care and reduced medication errors. However, they are still facing problems with e-prescribing and errors in e-prescriptions. The most common source of errors on e-prescriptions were due to inaccurate quantity of the medication, missing information, and unclear instructions of use. Wrong medication strength and dosage forms were also commonly reported on e-prescriptions.	Since e-prescribing became mandated by law to be used in all the pharmacies in Finland, the pharmacists thought that it helped with reducing medication errors and enhanced the dispensing process. However, there are still some ambiguities that continued with e-prescriptions that might lead to potential risks to the patients.
Lander [18]	Barriers to Electronic Prescribing: Nebraska Pharmacists’ Perspective	The objective of this study was to have a better understanding of the barriers that are causing delayed participation in using e-prescribing systems in pharmacies in the state of Nebraska. Another aim was to identify the impact of non-adoption among the pharmacists on the physicians’ ability to meet the expectations of meaningful use.	A total of 37 community pharmacies that did not accept e-prescriptions were targeted to participate in the study, 30 of which were eligible to participate. A total of 23 pharmacies agreed to participate, representing 77% of the target sample.	A qualitative study design.	Structured telephone interviews.	A total of 39% of the participants indicated that they were not planning on using e-prescriptions in the future, compared to 43% who were interested in e-prescribing sometime in the future. The main reasons identified as barriers to the implementation of e-prescribing in the community pharmacies included initial setup cost, negative impact on productivity during the implementation, transaction fees and maintenance costs, lack of interest from the patients or prescribers, low prescription volume to justify efficiency gains from e-prescribing, access to network connectivity or expense, current e-prescribing systems dissatisfaction, and limited awareness of the benefits of e-prescribing systems and their implementation procedure.	Identifying the barriers and drivers to the implementation of e-prescribing systems in the community pharmacies is a key element for successful adoption. In the US, some rural or small-size pharmacies are not participating due to financial concerns that have a direct impact on the sustainability of their pharmacies as a running business. Some financial considerations need to be addressed to increase the adoption of e-prescribing especially for rural pharmacies.
Farghali [6]	Pharmacist’s perception of the impact of electronic prescribing on medication errors and productivity in community pharmacies	Evaluating the perception of the Canadian community pharmacists of the impact of e-prescribing systems on the rates of medication errors and if they had any effect on the workflow or productivity in their pharmacies.	A total of 450 community pharmacists from different provinces across Canada that included Quebec, Ontario, Saskatchewan, Alberta, and British Columbia.	Secondary analysis of a national pharmacists’ survey.	Web-based survey through the Canadian Pharmacy Association and Canada Health Infoway that targeted community pharmacists across all the provinces.	Most of the pharmacists indicated that e-prescribing systems would have a positive impact on medication errors (66%) and increase the productivity in their pharmacies (70%). The community pharmacists thought that e-prescriptions would support their practice and would have positive outcomes on their patients’ care. However, the proportion of prescriptions received electronically remains very low compared to paper and faxed prescriptions.	The community pharmacists in Canada demonstrated their preparedness to utilize and work with e-prescribing systems to overcome the potential transcribing errors. The general perception was that e-prescribing systems would help the community pharmacists to have more efficient workflows. In Canada, the rates of e-prescriptions across the country remains low compared to other countries.
Gilligan [19]	Analysis of pharmacists’ interventions on electronic versus traditional prescriptions in two community pharmacies	The study aimed to measure the frequency of problems associated with e-prescriptions that required a pharmacist intervention to resolve them. The different types of problems and their frequencies were measured and assessed, as well as the time spent by the pharmacy staff to resolve these issues.	Two community pharmacies in the state of Arizona that belonged to the same chain grocery store and that were dispensing at least 100 prescriptions per day, 5% of which were e-prescriptions.	An observational prospective design.	Direct observation by four trained pharmacists as observers using the medication therapy intervention form created by Warholak and Rupp.	During the study period, around 9% (n = 153) of the new prescriptions (n = 1678) reviewed by the pharmacists required interventions. Analysis of the study results indicated that the rates of intervention were significantly different between e-prescriptions (11.7%) compared to faxed (3.9%) and verbal (5.1%) prescriptions.	The number of pharmacists’ interventions did not change with e-prescriptions compared to the other prescriptions formats. There are few potential opportunities for enhancing e-prescribing systems to reduce the errors identified on e-prescriptions. Even though e-prescribing systems might have solved some problems with the traditional paper prescriptions, new errors occurred that required the pharmacists to intervene and solve to complete the dispensing process of the medications.
Grossman [20]	Transmitting and processing electronic prescriptions: experiences of physician practices and pharmacies	The aim of the study was to explore the pharmacists and primary care physicians’ experiences with e-prescribing to evaluate the barriers and facilitators for implementing the system and using the direct communication and electronic prescription renewal features.	Community pharmacies and physician practices that are registered with Surescripts and participated in 12 community tracking study (CTS) sites were targeted. Out of which, 48 community pharmacies, 3 mail-order pharmacies, and 24 physician practices participated in the study. The pharmacist in charge in each pharmacy was the one interviewed in the participating pharmacies.	A qualitative study design.	Semi-structured telephone interviews.	Both physicians and pharmacists thought that e-prescribing can benefit their practices. Most of the participating physicians estimated that they sent 70% of their prescriptions electronically to the pharmacies. On the other side, the pharmacists estimated that less than 15% of the prescriptions they received were e-prescriptions. The process of sending of new e-prescriptions was satisfactory for the physicians and pharmacists. However, delays were often reported due to lack of training on the pharmacy side on identifying new e-prescriptions or sending the prescriptions to the wrong pharmacy. The prescription renewal was not utilized as expected because the renewal process was not always successful and to avoid the transaction fees. Processing e-prescriptions was not as efficient. Manual data input on e-prescriptions was still required for different reasons, like matching the right patient, drug name, strength, dosage form, quantity, and patient instructions.	e-Prescribing systems have the potential in improving the practice of both physicians and pharmacists, which will reflect on the positive outcome to the patients as well. The design of e-prescribing systems still have some challenges that impacted their efficient use in the physician offices and community pharmacies. Barriers to adoption and full-system usage need to be researched further to enhance the overall perception about the system. Some features of e-prescribing are still under-utilized and require more training and support to be provided.
Odukoya [21]	Relationship between e-prescriptions and community pharmacy workflow	The purpose of the study was to explore the retail pharmacy staff perceptions of the pros and cons of e-prescribing systems in their practices. The goal was to apply a sociotechnical framework to understand the impact of the system design on efficiency and safety of prescriptions processing in community pharmacies.	A total of seven retail pharmacies that included seven pharmacists and nine pharmacy technicians participated in the study.	An observational study design.	Direct observations and think aloud protocols.	Applying the STS theory revealed that the STS interactions included three constructs which are technology, people, and tasks. The design of the different e-prescribing systems included in the study had an impact on the pharmacists’ performance and the process of dispensing prescriptions. The main drivers for e-prescriptions included the consistency in presenting the prescription information, the elimination of legibility problems of paper prescriptions, and the time savings of archiving and retrieving prescriptions. The design flaws of e-prescribing systems included discrepancies between the prescriber and pharmacy computer systems that led to the inability to see the complete prescription information (drug names and instructions), patients and prescribers identification challenges, wrong drug quantities or dosage forms, and the inability to discontinue and identify similar prescriptions for the new orders.	The design of e-prescribing systems has a big impact on the safety of prescription medication dispensing process and the efficiency of the workflow in the retail pharmacies. The current system designs and implementation techniques may result in unanticipated risks that could potentially lead to medication errors and harm to patients if not resolved. More work needs to be completed in redesigning e-prescribing systems to achieve the best clinical outcomes to the patients.
Rahimi [22]	Pharmacists’ views on integrated electronic prescribing systems: associations between usefulness, pharmacological safety, and barriers to technology use	More than 25 million e-prescriptions are processed every year in Sweden. The study aimed to assess the outcomes of introducing the national Integrated Electronic Prescribing Systems (IEPSs) and its impact on efficiency and medication safety in retail pharmacies based on the Technology Acceptance Model (TAM).	A total of 52 out of 74 pharmacists in a Swedish municipality participated by returning their completed questionnaires with response rate of 70%.	A cross-sectional study design.	Postal questionnaire.	TAM was applied and modified to present the results of the study which demonstrated that most of the pharmacists perceived the IEPS to be useful and faster in processing prescriptions compared to the paper format. Other features of the system that were identified as useful included the ability of the system to reduce follow-up calls due to missing information or prescription ambiguity. Ease of use of the system was demonstrated by the impact on efficacy where the pharmacists thought that the IEPS saved time, was easier to access the IEPS, and was easier to interact with paper. Most of the pharmacists agreed that the IEPS was useful in reducing medication errors and improving medication safety. The barriers identified that might impact the system adoption included waste of time due to technical problems, and users’ stress due to their sense of becoming technology-driven in their practice.	The pharmacists’ perceptions indicated that the e-prescribing system improved their overall job performance. The prescription processing time has been reduced and the system was easier to interact with, compared to paper prescriptions. Patients’ safety was also enhanced, and medication errors were reduced. However, there are still technical challenges that could affect the system adoption and more efforts should be directed into maintenance and support.

**Table 4 healthcare-12-01280-t004:** Summary of e-prescriptions related variables impacting medication errors across the articles.

Incomplete Information	Clinical Issues
Missing information Quantity missing Patient not found Prescriber not identified	Drug–drug interaction Wrong drug Inaccurate dose Inappropriate quantity Inaccurate or incomplete instructions Inappropriate dosage form Use of additional notes field for the instructions

**Table 5 healthcare-12-01280-t005:** Summary of e-prescriptions related variables impacting productivity in pharmacy across the articles.

Missing information causing delays (for example, brand name necessary note) Need to edit the prescription instructions (sig) Missing prescription/sent to wrong pharmacy e-prescription not sent/delayed Violating legislations (sending controlled prescriptions via e-prescriptions) Duplicate prescriptions sent via e-prescription and other means (for example, fax, phone) Patient/prescriber information not found Indistinguishable e-prescription format Underuse of electronic renewal option

## Data Availability

No new data were created or analyzed in this study. Data sharing is not applicable to this article.

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
