# Peer review of "A Preliminary Scoping Review of the Impact of e-Prescribing on Pharmacists in Community Pharmacies"

_healthcare, 2024, doi:10.3390/healthcare12131280_

Round 1

Reviewer 1 Report

Comments and Suggestions for Authors

Thank you for the opportunity to review this manuscript. I feel some editing needs to be completed prior to publication.

Lines 53-59: definitions should be provided the first time the word is used, needs to be moved up to the start of the paper

Lines 60-63: this paragraph has nothing to do with the introduction. If it is needed, there needs to be a transition or explanation of why you are bringing up the types of pharmacies

Lines 91-94: The research questions don't seem to fit the rest of the paper as this is the first time that pharmacists' perceptions have been brought up, there is a difference between actual errors and how someone perceives them a situation. Is there a reason "perception" was not used as a search term?

Lines 114-123: seemed to repeat a lot of what was already said during section 4.2

Line 136: table numbering is out of order

Line 146: results section is too long - need to separate results from discussion (as currently there is not a discussion section)

Line 164: need citations about hospital pharmacies and research

Figures 2 and 3 seem unnecessary, not enough data to need the visualization. Text is enough.

Line 169-170: not sure if the word "participants" is the best option. Perhaps "subjects in the studies" or something else? "All participants in this review" - I think different wording is needed to clarify

Lines 181-182: why did you eliminate e-Prescriptions that were then printed out? I think those should factor into the issues you are looking for regarding workflow as that creates extra work and inefficiencies

Lines 249-250: don't understand how e-Prescriptions can control medication hoarding

Line 288-289: I disagree with technology's impact on workflow and medication safety being scarce. Need citation if you are going to say that

Line 303: need citation

Line 334: first mention of how it affects patients - would be helpful to bring up sooner, including in the introduction

Line 374: since only 4 countries are included, not sure you can say these results are generalizable

Table 5: too hard to read in current format - needs edited or else included as an appendix with an abbreviated table included in the actual article

Comments on the Quality of English Language

English looked appropriate

Reviewer 2 Report

Comments and Suggestions for Authors

I recommend publishing the research for its high impact on the significance if using automation and information technology in pharmacy practice, either in community pharmacy or clinical pharmacy, to maximize efficacy and safety and minimize medication errors.

Author Response

Thank you very much for taking the time to review our paper.

Reviewer 3 Report

Comments and Suggestions for Authors

Electronic prescribing (e-Prescribing) is intended to establish a direct communication between the prescriber and pharmacy computer systems. One of the major benefits promoted for the adoption and implementation of e-Prescribing is the elimination of medication errors due to illegible hand-written prescriptions, which should enhance the quality and safety of the prescribing stage of the medication use process and ultimately improve patient outcomes. This scoping review aimed to map the available literature and provide an overview of the published articles discussing the impact of e-Prescribing on pharmacist work, workflow, medication errors, productivity, and patient outcomes in community pharmacies.

The reviewer offers the following comments, by section of the manuscript, for the authors’ consideration.

Abstract

For the Objective of the Abstract (lines 8-10), the authors indicated that this scoping review aims to “provide an overview of the published articles discussing the impact of electronic prescribing on medication errors and pharmacy workflow.” While medication errors and pharmacy workflow are noted as endpoints in the title of the manuscript, the vexing question is: What about the other endpoints such as productivity and patient outcomes?

Introduction

In the first sentence of the second paragraph of this section (page 2, lines 46-47), the authors noted that several studies have suggested multiple benefits associated with e-Prescribing, yet the authors only cited one published article and omitted additional context such as the care setting. It is worth noting that the cited article (i.e., Porterfield et al., 2014) was a systematic review aimed to explore the benefits that e-Prescribing has had in improving the efficacy, accuracy, and cost of prescribing in ambulatory care settings and to assess the barriers to its implementation, at an earlier period in the history of the adoption and implementation of e-Prescribing. This is worth clarifying in the context of the present scoping review. For instance, the authors might consider rewriting this sentence to read: “According to a systematic review of e-Prescribing practices in the ambulatory care setting published about a decade ago, e-Prescribing reduces prescribing errors, increases efficiency, and helps to save on healthcare costs.” Furthermore, perhaps this sentence is better suited as the first sentence of the last paragraph of this section, whereby the authors briefly discuss the difference between community and hospital pharmacies (i.e., ambulatory and non-ambulatory care settings).

As a follow-up to the previous comment, the information provided in the second paragraph about the definition of prescribing errors (page 2, lines 47-52) is arguably superfluous and distracts from the study rationale. As such, the authors should consider omitting this information. This could help improve the flow of the Introduction (e.g., intentions of e-Prescribing, descriptions of e-Prescribing, definitions of e-Prescribing). Nevertheless, the flow of the Introduction might further be improved by reframing the problem. In doing so, the authors might consider answering the following questions: What is e-Prescribing? What are the intentions of e-Prescribing, including its major benefits? What are the potential pitfalls of e-Prescribing? How does e-Prescribing in the community setting differ from e-Prescribing in the hospital setting and how do they intersect? Why is it important to study e-Prescribing in the community setting, in particular?

Study Rationale

While awareness of the impact of technology changes on pharmacy work environments and patient outcomes indeed needs to be raised, there is a glaring omission of a description of previous studies that have been published on this topic, most notably the aforementioned systematic review by Porterfield et al. This section would benefit from a succinct summary of what is already known on this topic, what gaps remain, and how this scoping review aims to address those gaps.

Study Objective

The authors stated that their objective was to provide an overview of the “community pharmacists’ perception” of the impact of e-Prescribing on various endpoints. One may surmise that “perception” implies qualitative, observational, or subjective research, such as questionnaires or surveys. Did the authors intend to only evaluate these types of research studies in their review? If not, then the authors should consider modifying their verbiage. In considering such modification, the authors might also assess how the objective stated in this section aligns (or does not align) with the objective stated in the Abstract. Additionally, the authors should consider that, later in the Methods (section 4.2), they indicated that they identified relevant literature, whether qualitative or quantitative studies (page 3, line 96).

Methods

In the first step of the scoping review methodology (i.e., section 4.1), the authors noted that their prior research on this topic yielded limited research that addressed the impact of e-Prescribing on community pharmacies (page 2, lines 88-89). To the reviewer’s previous points, this statement is probably better suited for the Introduction section, along with a brief description of the authors’ findings. Doing so would better frame the problem(s) for the reader and thereby establish the context of the research questions for this scoping review.

In section 4.2 (page 3), it might be helpful to describe the Grey Literature Report very briefly to ensure that the readership is familiar with this database.

Also, in section 4.2 (page 3), was the literature search limited to articles that evaluated e-Prescribing in the United States? This might be worth mentioning, along with a statement of the rationale for choosing only US-based studies or including studies beyond just the United States.

Considering that “study selection” was the third step in the methodological framework for this scoping review, should this subsection of the Methods be labeled as 4.3 instead of 5 (page 3)? If so, then conceivably the two steps that follow (i.e., Charting the Data and Collating, Summarizing, and Reporting the Results) should be labeled as subsections 4.4 and 4.5, respectively (page 4).

In the Study Selection subsection (page 3), it appears that the majority of the first paragraph (lines 116-123) is a repeat of the information provided in section 4.2 (i.e., Identifying Relevant Studies). Please clarify in the text or consider moving some of the non-repetitive information to section 4.2; otherwise, consider omitting this repetitious information.

In the Study Selection subsection (pages 3-4), the researchers indicate that both literature and systematic reviews were excluded from their study (Table 2). Subsequently, in the Results section (page 4, lines 150-151), the researchers specify that 30 articles were found by searching the Cochrane Database of Systematic Reviews®. This begs the questions: What was the purpose of searching this database if both literature and systematic reviews were to be excluded a priori? Where is a noted how many articles found from the Cochrane Database of Systematic Reviews® search were excluded (e.g., Figure 1)?

In the Charting the Data subsection, why does Table 5 come / get mentioned before both Table 3 and Table 4? Should this table be renumbered or perhaps not mentioned until the Results section?

Results

Figure 1 illustrates a flow diagram for the study selection process. According to the diagram, 24 full-text articles were assessed by the researchers for eligibility. Among those screened, 14 articles were excluded, including 8 articles that dealt with e-Prescribing in the non-community pharmacy setting and 4 editorials. With that said, the authors only listed reasons for excluding 12 articles that were screened, not 14 as stated in the diagram. This begs the question: What was the reason(s) for excluding the 2 other articles? Please ensure that all excluded articles have been accounted for in the diagram.

Regarding the Study Characteristics, the authors noted that “the focus of most of the literature remains on hospital pharmacies” (page 5, lines 162-163). By this reviewer’s count, the researchers excluded 8 articles that focused on e-Prescribing in the non-community pharmacy setting, which presumably indicated hospital pharmacies. On the other hand, the researchers included 10 articles that focused on e-Prescribing in the community pharmacy setting. Therefore, how do the authors substantiate their claim that most of the searched literature focused on hospital pharmacies? Please explain in your response to the journal and/or clarify in the manuscript text.

Considering that there was no mention of this scoping review entailing a time delineation or publication year analysis, is Figure 3 necessary? The authors might consider asking themselves: What value, if any, do the data in Figure 3 add to this manuscript?

While Table 4 is mentioned in the Methods section, it is not mentioned (again) in the Results section. In general, a table that presents study results should be mentioned in the Results section. Please consider referencing Table 4 in the e-Prescribing Impact on Productivity subsection of the Results, similar to how Table 3 is referenced in the e-Prescribing Impact on Medication Errors subsection (page 6, line 199).

Table 5 provides a summary of articles included in this review. However, this table appears to only include a summary of 8 articles, yet the researchers included 10 articles in this review. Please account for the other 2 articles that were not included in Table 5.

Discussion

Similar to an earlier comment, the authors’ claim that the number of publications on e-Prescribing in hospital pharmacies (and primary care settings) continues to outnumber those in community pharmacies (page 9, lines 289-291) does not appear to be substantiated based on the information provided in this manuscript. Please provide clarifying information in the text to elucidate this claim.

In general, the Discussion section is largely a redaction of the researchers’ findings from the Results section. The Discussion could be enhanced by describing how these findings potentially translate into economic, clinical, and humanistic outcomes in the broader context of medication errors and the medication use process in community pharmacy settings. Additionally, did the researchers find or conceive any potential benefits of e-Prescribing? As currently constructed, the Discussion focuses only on the negative aspects of e-Prescribing in community pharmacy settings. Lastly, might the authors provide insights into gaps that still need to be addressed, including study design methodologies and future research endeavors?

In the first two sentences of the opening paragraph of the Limitations section (page 10, lines 368-371), the authors write: “This scoping review is first attempt to provide evidence of and highlight the impact of e-Prescribing systems upon community pharmacies. To the best of our knowledge, no other reviews focused on community pharmacy workflow and error in the context of e-Prescribing systems.” Is this true? What about the systematic review article published by Porterfield et al. in 2014, which is reference No. 5 cited by the authors? Additionally, what about any of the 30 additional articles – systematic reviews or scoping reviews – that were found from the researchers’ search of the Cochrane Database of Systematic Reviews®?

In the Limitations section (page 10, lines 371-375), the authors note that one of the strengths of this scoping review was the inclusion of articles from differing countries, which helped to identify common e-Prescribing challenges experienced by varying nations globally. While the inclusion of a few articles from countries beyond the United States certainly strengthens the generalizability of this review’s findings, arguably, the researchers did not provide a comparative analysis on this matter, comparing the similarities and differences of the impact of e-Prescribing across community pharmacies from different countries. Perhaps this claim could be “softened” to suggest that this review shed insights on this topic. but more research is necessary; otherwise, the authors should consider omitting this claim.

Conclusion

The second paragraph of the Conclusion section (page 11) introduces a relatively new topic that was not vetted out in the body of the manuscript, namely the potential benefits of implementing e-Prescribing systems in community pharmacies. In general, the Conclusion should succinctly summarize the researchers’ interpretation of their main findings and, as such, should not introduce new topics. Conceivably, there are two solutions to rectify this problem: (1) the authors could present and discuss results of the potential benefits of implementing e-Prescribing systems in community pharmacies in the body of the manuscript, or (2) the authors could omit discussing these potential benefits in the Conclusion section.

Comments on the Quality of English Language

There are many instances throughout the manuscript where grammar, punctuation, and tense could be improved. As one example, in the second paragraph of the Discussion, the authors wrote: "The literature suggests that e-Prescribing helped reducing the risk of medication errors contributed to poor legibility of handwritten prescriptions, decreasing the turnaround time for refills requests, and call-backs for unsigned prescriptions." As another example, the first sentence of the Limitations section reads: "This scoping review is first attempt to provide..." These are just two examples for your consideration.

Reviewer 4 Report

Comments and Suggestions for Authors

Dear authors,

I would like to thank the journal for providing me with the opportunity to review your manuscript titled "The Impact of Electronic Prescribing on Pharmacist Work, Workflow, Medication Errors, Productivity, and Patient Outcomes in Community Pharmacies: A Scoping Review." The topic represents one of the most significant challenges, and I commend you for studying it in such depth.

Firstly, I request that you review the text formatting as the references, for example, do not adhere to the criteria used by MDPI.

To facilitate your work, I will divide my comments as follows:

  • Abstract: It is necessary to rewrite it in a way that better highlights your results.
  • Introduction: The background section is too verbose; you need to be more concise. Additionally, I ask you to merge the parts "Introduction," "Study Rationale," and "Study Objective" into a single paragraph.
  • Methods: They are fairly clear, but it is important to merge "Methods" and "Study Selection" as they constitute a single section. I also request the removal of Figure 2 and Figure 3 as they do not add anything to the text.
  • Results: They are quite clear.
  • Discussion: Lines 348-355 are not sufficiently clear. Furthermore, I ask you to expand upon what was discussed in 356-362 with the analysis provided in 10.1136/bmjopen-2022-065301 and 10.1136/bmj.m1822. Additionally, please pay attention to Table 5; it needs to be rewritten more clearly.

I hope these suggestions will be helpful for the improvement of your manuscript.

Best regards

Comments on the Quality of English Language

Moderate editing of English language is required

Round 2

Reviewer 1 Report

Comments and Suggestions for Authors

Thank you for the opportunity to review your revised manuscript. I think the changes made have improved the flow and understanding of the paper. I appreciate the thoughtful responses to previous suggestions. Congratulations and I look forward to seeing this manuscript published!

Author Response

Thank you very much for your feedback and support!

Hope you have a wonderful day

Reviewer 3 Report

Comments and Suggestions for Authors

Thank you for adequately addressing my comments and suggestions.

Comments on the Quality of English Language

The quality of writing has been improved.

Author Response

(The authors gave the same response as above.)

Reviewer 4 Report

Comments and Suggestions for Authors

Dear Authors,

I appreciate the opportunity afforded by the journal to review this manuscript for the second time. Upon reviewing the revisions made, I acknowledge a slight improvement, albeit insufficient to meet the required standards.

I urge you to seriously consider the suggestions provided in my previous reviews, as failure to address them may render the article unpublishable. Specifically, the manuscript must be structured according to the sections outlined for review articles, as outlined in the MDPI guidelines. Familiarizing yourselves with these guidelines is crucial in ensuring compliance.

Furthermore, I reiterate my previous comment regarding the lack of clarity in lines 348-355. Additionally, I urge you to elaborate on the discussion presented in lines 356-362, drawing upon the analyses provided in references 10.1136/bmjopen-2022-065301 and 10.1136/bmj.m1822.

I trust that you will carefully consider these suggestions and make the necessary revisions to enhance the quality and coherence of the manuscript. Your attention to these matters is essential in ensuring the publication readiness of the article.
